# Individual-Level Comparisons of Honey Bee (Hymenoptera: Apoidea) Hygienic Behavior Towards Brood Infested with *Varroa destructor* (Parasitiformes: Varroidae) or *Tropilaelaps mercedesae* (Mesostigmata: Laelapidae)

**DOI:** 10.3390/insects11080510

**Published:** 2020-08-07

**Authors:** Monica Shrestha, Jakob Wegener, Ishan Gautam, Madhusudan Singh, Christoph Schwekendiek, Kaspar Bienefeld

**Affiliations:** 1Department of Breeding and Genetics, Bee Research Institute, 16540 Hohen Neuendorf, Germany; monica_2762@yahoo.com (M.S.); schwekec@hu-berlin.de (C.S.); kaspar.bienefeld@hu-berlin.de (K.B.); 2Natural History Museum of Tribhuvan University, Kimdol Manjushree Bazar, Kathmandu, Nepal; is_gautam@rediffmail.com (I.G.); madhusudanmansingh@gmail.com (M.S.)

**Keywords:** parasitic mites, artificial infestation, brood removal

## Abstract

**Simple summary:**

Parasitic mites are among the main causes of honeybee colony losses worldwide. Long-term selection has led to bees that are, in part, resistant to one specific mite, *Varroa destructor*. We investigated whether an important Varroa resistance trait, “hygienic behavior”, is also effective against *Tropilaelaps mercedesae*. *T. mercedesae* is another bee-parasitic mite of great economic importance and a lifecycle similar to that of *V. destructor*. “Hygienic behavior” means the ability of bees to recognize and destroy mite-infested bee brood, interrupting reproduction of the parasite. We also compared the expression of the behavior in two species of bees, one of which (*Apis cerana*) is thought to be resistant to *T. mercedesae*, while the other (*Apis mellifera*) is sensitive to it. We confirmed that both bees were able to express hygienic behavior also towards *T. mercedesae*. Moreover, we found that specialist bees destroying *V. destructor*-infested brood often also destroyed brood infested by *T. mercedesae*. This strongly suggests that hygienic behavior against the two mites is really the same trait, and selecting for *V. destructor* resistance likely also increases resistance to *T. mercedesae*. We also showed that *A. cerana* is more efficient at destroying *T. mercedesae*-infested brood, which may help to explain its resistance.

**Abstract:**

The mites *Varroa destructor* Anderson and Trueman and *Tropilaelaps mercedesae* Anderson and Morgan are both serious threats to the *Apis mellifera* beekeeping industry. A trait frequently used in selection programs for *V. destructor* resistance is hygienic behavior, the selective removal of diseased/damaged brood. Here, we measured the level of association of the expression of hygienic behavior against both mites in *A. mellifera*, by observing whether the same individual bees would carry out the opening and removal of brood infested by the two parasites. The groups of bees showing these behaviors on cells artificially infested by either parasite showed a large overlap, making it appear likely that the two traits are at least closely coupled. Therefore, breeding for *V. destructor* resistance based on hygienic behavior could prepare *A. mellifera* populations for dealing with *Tropilaelaps* sp. mites, and vice versa. Using the same bioassay, we also compared the hygienic behavior of *A. mellifera* towards *T. mercedesae* to that of the Asiatic honey bee, *Apis cerana*. *A. cerana* workers eliminated a greater proportion of infested cells, which may in part explain the resistance of this bee to *Tropilaelaps* and the observation that *Tropilaelaps* reproduction on brood of this species is extremely rare.

## 1. Introduction

Parasitic mites and associated pathogens are among the most frequent causes of *Apis mellifera* (L.) colony losses in most parts of the world [1,2,3,4]. In Europe, North and Central America, and most of temperate Asia, *Varroa destructor* (Anderson and Trueman) is by far the mite species with the greatest economic impact. In those parts of Asia in which the introduced *A. mellifera* coexists with autochthonous giant honey bees (*Apis dorsata*, *Apis breviligula*, and/or *Apis laboriosa*), mites of the genus *Tropilaelaps* are often equally or even more damaging to the *A. mellifera* beekeeping industry [5,6].

The Asian giant honey bees are thought to be the original hosts of the genus *Tropilaelaps* [5]. Like *V. destructor*, all four *Tropilaelaps* species reproduce inside capped brood. Two species, *Tropilaelaps clareae* and *Tropilaelaps mercedesae*, are known to cause damage also to *A. mellifera* colonies, and at least *T. mercedesae* (Anderson and Morgan) is capable of transmitting Deformed Wing Virus as well as black queen cell virus [7,8]. Unlike *V. destructor*, adults of *T. mercedesae* also feed on early larval stages of *A. mellifera* [6]. It is not clear, however, whether mites of the genus *Tropilaelaps* are able to feed on adult bees, and they survive for only a few days if the host colony is broodless [5]. This is thought to be an important limiting factor of the spread of this genus of bee parasitic mites. However, the main condition for *T. mercedesae* survival, continuous brood rearing, is fulfilled in many parts of the distribution area of the western honey bee, and there are indications that the mite may also survive on rats [9]. Therefore, tropilaelapidosis is seen as a serious risk to *A. mellifera* beekeeping worldwide [10].

The best-studied Asian honey bee, *Apis cerana* (Fab.), appears to be resistant to both mites of the genus *Tropilaelaps* and *V. destructor* [11,12,13]. In the case of *V. destructor*, this resistance is probably due to a combination of behaviors of the adult bees, including grooming, “entombing” of infested drone pupae, and removal of infested worker brood, with the inability of the mite to reproduce on worker brood [11,14,15]. In the case of *T. mercedesae*, grooming behavior seems to play an important role [16]. Whether *A. cerana* removes *T. mercedesae*-infested brood has not yet been tested to our knowledge.

Beekeepers working with *A. mellifera* in regions where either *T. mercedesae* or *T. clareae* occur have to find ways to control these parasites, since untreated *Tropilaelaps* infestations regularly lead to colony collapse in this honey bee species. Although chemical treatments used to control *V. destructor* seem, for the most part, also effective against mites of the genus *Tropilaelaps* [17,18,19], the availability of resistant stocks of bees would represent a more sustainable solution. In the case of *V. destructor*, more than four decades of selective breeding, as well as natural selection of resistance traits in isolated populations, are finally beginning to show results [20,21,22,23,24]. Hygienic behavior, the detection and removal of damaged/diseased brood by worker bees, affects mite reproductive success [15] and is central to many resistance breeding programs. Khongphinitbunjong et al. [25] have studied *T. mercedesae* non-reproduction and infested brood removal by *A. mellifera* worker bees in Thailand, where bees of this species have been exposed to *T. mercedesae* for many years. They found removal rates of artificially infested brood of 53%, and high rates of non-reproduction on artificially and naturally infested *A. mellifera* brood. This seems to indicate that in *A. mellifera*, hygienic behavior could be a promising resistance trait not only with regard to varroosis, but also to tropilaelapidosis.

An important question now is whether the removal of *T. mercedesae-* and *V. destructor*-infested brood really constitutes the same trait. Given that the type of injuries produced by feeding of *T. mercedesae* on larvae and pupae is distinct from that produced by *V. destructor* [6,26], it may be hypothesized that different genetic specializations are required on behalf of the hygienic workers in order to sense them across the cell capping. Also, the role of hygienic behavior in *A. cerana* resistance to *T. mercedesae* remains to be elucidated.

In the present study, we aimed to:(1)Compare hygienic behavior of *A. mellifera* and *A. cerana* against *T. mercedesae*. For this, we first observed hygienic behavior separately in monospecific groups of bees. We then went on to observe mixed groups of bees on brood of either species. This was done with the objective of comparing their reaction to strictly identical removal stimuli.(2)Verify whether hygienic behavior against *T. mercedesae* and *V. destructor* are co-expressed in *A. mellifera*, by determining whether the same individual worker bees perform these behaviors.

## 2. Material and Methods

### 2.1. Bees and Mites

The experiments were performed in an experimental apiary in Kathmandu, Nepal. Colonies of *A. mellifera* used in the study included local hybrids as well as hives headed by *Apis mellifera carnica* queens imported from Germany. *A. cerana* was represented by colonies from three Nepalese regions, the Terai, the Kathmandu Valley, and the mountains around this valley. Bees from the different stocks within both species were mixed in the experiments, equalizing the numbers of bees from each stock as far as possible. All mites were obtained from local *A. mellifera* colonies, and were collected from newly-capped brood (prepupal or white-eyed stages). Great care was taken not to injure the brood while collecting the mites, and also in order to avoid contamination through leaking body fluids. The species affiliation of *V. destructor* and *T. mercedesae* was verified morphometrically by applying published identification keys [12,27].

### 2.2. Bioassay for Observing Hygienic Behavior

The bioassay used is described in detail in [28] and is depicted in Figure 1. The day before an experiment, combs of emerging brood were removed from the source colonies. An additional comb of freshly-capped brood was taken from a donor colony, unrelated to the bees used in the experiment, to be used for artificial infestation. Source colonies of these combs were treated with Amitraz, and were monitored for infestation by opening variable numbers of brood cells in order to avoid using brood with sizeable natural infestation. Between 20 and 35 mites were introduced into brood cells with a brush (one per cell) after carefully opening the cells with a razor blade. The caps were closed again, and their positions marked on transparency sheets. Between three and five cells were opened and closed without the introduction of a mite (“sham-manipulation”), and served to check whether brood removal was due to the presence of the mite or to manipulation of the cell. The comb was then placed inside a hive previously shown to present little brood hygiene, for repair of the manipulated cell caps. Groups of 1100–2100 newly-emerged (<24-h-old) worker bees were individually labelled with opalith plates of different shapes and orientation, using nail polish (Maybelline Express; L’Oréal, Paris, France) as glue. After 24 h, the manipulated comb was placed in a wire mesh cage with one glass side. The labelled bees were introduced together with a mated queen, and the cage was placed inside a strong colony of the same species as the labelled bees, with the glass pane facing an opening in the side wall of the hive. In cases where a mix of *cerana* and *mellifera* workers were used, the species of the fostering colony was alternated between repetitions of the experiment. Hygienic behavior was then observed under infrared illumination with the help of a suitable CCTV camera (PS/DX4-285GE; Kappa optronics, Gleichen, Germany). Observations were recorded continuously for 4–5 days (corresponding to days 2 to 5–6 post-infestation), and later viewed. From these videos, we determined the time of cell openings, as well as the identity of the bee initiating the cell opening, and of the first two bees continuing with this task. In order to limit the effort required to analyze the ~2400 h of video material, we only looked at those cells that had been emptied at the end of the experiment. Only the event that ultimately led to the removal of the pupa was observed. Potential earlier attempts at uncapping and events of recapping could not be included. The intensity of these behaviors was judged as either “low” (a bee nibbling briefly at a cell cap and then leaving), “medium” (a bee nibbling repeatedly with brief interruptions or for a period of up to 30 s), or “high” (sustained nibbling for >30 s).

### 2.3. Comparisons

The above-described assay was performed 19 times, with a total of 35,910 tagged worker bees:(1)Five times with *A. cerana* bees on *A. cerana* brood (designated below as “cbees_cbrood”).(2)Four times with *A. mellifera* bees on *A. mellifera* brood (“mbees_mbrood”).(3)Eight times with equal numbers of bees from both species and alternating origins of brood (“xbees_xbrood”).(4)One time with *A. cerana* bees on *A. mellifera* brood (“cbees_mbrood”).

One round of the experiment could not be used, because bees started to indiscriminately remove all brood, possibly due to overheating. *V. destructor* and *T. mercedesae* mites were used together in the same experiments at a ratio of approximately 1:2. In four of the experiments with *A. cerana* bees on *A. cerana* brood, only *T. mercedesae* mites could be used because no *V. destructor* mites were available.

### 2.4. Data Analysis

All statistical analyses were conducted using IBM SPSS software (IBM, Amonk, NY, USA).

We first verified, for each of the bees/brood combinations cbees_cbrood, mbees_mbrood, and xbees_xbrood, whether hygienic behavior was specific to infested cells, i.e., whether the bioassay really allowed us to observe mite-related responses. We did this by comparing removal ratios using the χ^2^ test. These tests were performed separately for *T. mercedesae*- and *V. destructor*-infested cells.

Next, we compared the removal dynamics of *Tropilaelaps*-infested brood among the three bee/brood combinations cbees_cbrood, mbees_mbrood, and xbees_xbrood. This was done using Kaplan–Meier analysis with pairwise comparisons (Mantel–Cox test). In the case of the experiments with mixed worker bees (xbees_xbrood), experiments with *A. mellifera* brood and with *A. cerana* brood were distinguished, in order to also visualize the role of the species of brood used.

We then went on to compare the efficiency and intensity of cell opening behavior towards *T. mercedesae*- and *V. destructor*-infested brood in *A. mellifera* using χ^2^ tests on the frequency distributions. Only data from those experiments in which bees and brood were of one and the same species were used for this analysis, because we could not exclude interactions between the factors “species of brood” and “species of workers”. In a separate analysis, we compared the intensity of opening events on *T. mercedesae*-infested brood from the mixed-species experiments (xbees_xbrood), contrasting cases in which workers had opened brood of their own or the other species.

Finally, we investigated whether a bee involved in the opening of cells infested by one of the two mite species had a greater-than-arbitrary probability of also being involved in the removal of the other. For this, we calculated the proportions of both *T. mercedesae*- and *V. destructor*-infested cell-openers in all bees in the experiments. We multiplied these two probabilities to obtain the likelihood of any bee to become an opener of both types of cells, under the hypothesis that the two traits occurred independently. This theoretical likelihood was then compared with the observed frequency of openers of both cell types using the χ^2^ test. This analysis was performed only for *A. mellifera* bees, because too few cases of *A. cerana* opening *V. destructor*-infested cells were available, due to the lack of *V. destructor* mites during experiments with this bee species.

The significance level of the type 1 error for all tests was 0.05.

## 3. Results

### 3.1. Specificity of Opening Behavior to Infested Cells

*A. mellifera* workers opened and removed *V. destructor*-infested brood significantly more frequently than sham-manipulated brood (45.5 vs. 8.8%; total *n* = 56; χ^2^ = 5.5; *p* = 0.017). The same was true for mixed-species groups of worker bees (xbees_xbrood; 43.6 vs. 18.2%; total *n* = 61; χ^2^ = 8.4; *p* = 0.004). In the cbees_cbrood experiments, insufficient numbers of cells were infested with *V. destructor* to carry out the test. Similar to *V. destructor*, the removal of *T. mercedesae*-infested larvae/pupae by *A. mellifera* workers was a specific response to infestation, rather than to the manipulation of the cells alone (37.6 vs. 8.8%; total *n* = 121; χ^2^ = 4.1; *p* = 0.043). This was also true for the xbees_xbrood experiments (41.3 vs. 18.2%; total n = 148; χ^2^ = 4.2; *p* = 0.039). The frequency of removal of *T. mercedesae*-infested brood by *A. cerana* (experiments cbees_cbrood), however, was not significantly different from that observed in sham-manipulated brood (37.6 vs. 20.0%; total *n* = 129; χ^2^ = 2.3; *p* = 0.13).

### 3.2. Comparison of Removal of T. mercedesae- and V. destructor-Infested Brood Cells by A. mellifera

Here, “survival” means that brood was not opened and removed by the worker bees. The graph is based on a total of 745 artificially infested cells. The period of 0–24 h post-infestation is not displayed, because combs were placed inside a non-hygienic hive for repair of manipulated cell caps by the bees. The time course of brood survival differed significantly between *T. mercedesae*- and *V. destructor*-infested cells (χ^2^ = 11.3; df = 1; *p* = 0.001).

The graphs are based on the following numbers of observations (= opening events; from left to right): 38; 49; 54; 22. In the case of *cerana* bees observed on *cerana* brood, no cells were infested with *V. destructor* mites. The intensity of opening behavior shown by *A. mellifera* bees was not different for *T. mercedesae*- and *V. destructor*-infested brood of this same species (left part of graph; χ^2^ = 3.9; *p* = 0.14). Different combinations of the factors “species of bees” and “species of brood” produced different intensities of opening behavior towards *Tropilaelaps*-infested brood (comparison of the four parts of the graph; χ^2^ = 14.9; *p* = 0.02); n.a.: not analyzed.

Figure 2 depicts the time course of infested brood removal of larvae/pupae for the two species of mites. Only *A. mellifera* bees observed on *A. mellifera* brood are included here. It shows that, although both types of infested brood are removed, *V. destructor*-infested cells are emptied slightly faster and to a greater proportion (log-rank test; χ^2^ = 11.3; df = 1; *p* = 0.001; total number of cells observed = 745). The intensity of removal behavior of *A. mellifera* towards *T. mercedesae*- and *V. destructor*-infested brood was not different, however (initiation of uncapping: χ^2^ = 3.9; total *n* = 49; *p* = 0.14; continuation of uncapping/removal: χ^2^ = 0.1; total *n* = 89; *p* = 0.98; data for initiation of uncapping shown in Figure 3). The chance of any *A. mellifera* worker bee to perform components of hygienic behavior on both *T. mercedesae*- and *V. destructor*-infested brood was far greater than would be expected if the two traits had occurred independently of each other (χ^2^ = 9.0; df = 1; total *n* = 1213; *p* = 0.003).

### 3.3. Comparison of Tropilaelaps Removal by A. mellifera, A. cerana, and Mixed-Species Groups of Worker Bees

Here, “survival” means that brood was not opened and removed by the worker bees. The graph is based on a total of 1228 artificially infested cells. The period of 0–24 h post-infestation is not displayed, because combs were placed inside a non-hygienic hive for repair of manipulated cell caps by the bees. The time course of brood survival differed significantly between treatments (χ^2^ = 26.8; *p* < 0.001).

Figure 4 compares the cumulated survival over time of *Tropilaelaps*-infested cells in the treatments mbees_mbrood, cbees_cbrood, and xbees_xbrood. It can be seen that *A. cerana* bees, when observed on their own brood and in the absence of *A. mellifera* workers, showed the fastest removal of infested brood, and also removed the greatest proportion of it. This difference was significant for the comparisons with mbees_mbrood, as well as with xbees_cbrood (χ^2^ = 20.7 and 7.3; *p* <0.001 and 0.007), while it was marginally non-significant in the case of xbees_mbrood (χ^2^ = 3.2; *p* = 0.075). The three sets of experiments in which *mellifera* workers were used, either alone or in combination with *A. cerana*, formed a homogeneous block (χ^2^ = 2.3–3.8; *p* = 0.127–0.053), indicating that the presence or absence of *A. cerana* workers did not influence hygienic behavior in these experiments. *A. cerana* and *A. mellifera* did not differ with regard to the median duration of uninterrupted cell opening events, although the range of durations was far wider in *A. cerana* (total *n* = 167; *A. cerana*: median = 104 s (7 to 5569); *A. mellifera*: median = 94 s (1 to 630)). The intensity of initiations of cell openings was generally quite similar in all treatments, but judged slightly lower in the case of xbees_cbrood than in the others (χ^2^ = 14.9; *n* = 169; *p* = 0.02; Figure 3). In contrast, the intensity of continuations of cell opening and brood removal was lowest in treatment mbees_mbrood (χ^2^ = 13.2; *n* = 262; *p* = 0.04; data not shown).

### 3.4. Hygienic Behavior in Mixed Groups of Worker Bees

Hygienic behavior in mixed groups of workers (experiments xbrood_xbees) was almost exclusively performed by *A. mellifera* bees, despite the fact that equal numbers of individuals from both species were present, and regardless of the species affiliation of the brood (173 observed actions of initiation or continuation by *A. mellifera* vs. only 4 by *A. cerana*). In order to clarify whether the absence of hygienic behavior by *A. cerana* on brood of *A. mellifera* was due to factors linked to the brood, we performed an additional experiment in which an *A. cerana*-only group of worker bees was placed on infested *A. mellifera* brood. In this experiment, 16 infested pupae out of 25 were removed.

The fact that, in each of the xbees_xbrood experiments, one half of the bees was confronted with brood from another species did not lead to indiscriminate removal of brood. This is illustrated by the removal of sham-manipulated cells, which was not greater in the “xbees_xbrood” experiments than in the mbees_mbrood or cbees_cbrood experiments (χ^2^ = 1.6; df = 2; total *n* = 86; *p* = 0.46).

## 4. Discussion

It has been reported several times that *T. mercedesae*, like *V. destructor*, can trigger hygienic behavior towards infested brood in *Apis mellifera* ([25,29], Ritter and Schneider-Ritter 1987; as cited in [15]). Our data confirm this finding. Khongphinitbunjong et al. [25] have also shown that *T. mercedesae* reproductive success in cells targeted by hygienic behavior is effectively reduced. However, while the earlier authors agree on the fact that cells singly infested by *T. mercedesae* females are removed faster than cells singly infested by *V. destructor*, our results point to the contrary. It may be speculated that differences between local strains of the two mite species are responsible. The two earlier studies were performed with locally-occurring mites in Thailand, while ours was performed in Nepal. According to Anderson et al. [30], at least two different haplotypes of *T. mercedesae* are present in Thailand, one of which is the “mainland Asia” type also found in both India and China, the two neighboring countries of Nepal. Samples of *V. destructor* from the Kathmandu region have been genotyped by Solignac et al. [31]. They presented an unusually high heterozygosity in microsatellite markers, and were classified as belonging to the rare “Japan” haplotype, based on the analysis of mtDNA. This haplotype is thought to show a lower virulence than the more common “Korean” haplotype [31]. Our bioassay involved the transfer of mites from early capped stages of donor brood to similar stages of recipient brood. As reproduction of both *V. destructor* and *T. mercedesae* started shortly after cell capping [26,32], the transfer might have interrupted reproduction, thereby influencing the onset of reproduction-related removal stimuli. From earlier experiments, we know that most transferred mites nevertheless manage to produce viable offspring. As a possible retardation of reproduction would concern both mites alike, it can be assumed to be of minor importance for the interpretation of the present results.

For more than 30 years now, big efforts have been, and are being, made to increase the prevalence of hygienic behavior (and related traits such as Varroa sensitive hygiene/suppressed mite reproduction) in populations of *A. mellifera* in many parts of the world [28,33,34,35,36,37,38,39]. If it can be shown that hygienic behavior against *V. destructor* and against *T. mercedesae* are really the same trait, or at least closely coupled, then this would mean that breeding for *V. destructor* resistance through selection on hygiene traits is also a preventive strategy in view of a possible introduction of *T. mercedesae*. It has been reported that lines of bees partially resistant to the latter (from a commercial strain used in Thailand) and bees partially resistant to *V. destructor* (ARS (Agricultural Research Service) “Russian bees”) are characterized by similarly high levels of hygienic behavior, as measured by the freeze-killed brood assay, making it likely that the genetic bases of *V. destructor*- and *T. mercedesae*-specific hygiene are at least linked [40]. Even in strongly hygienic colonies, recognition and removal of damaged brood is a behavioral specialization only expressed by a small minority of the worker bees [28]. Our data now show that *V. destructor*- and *T. mercedesae*-infested cells are often opened by the same specialized individuals, meaning that the two traits were coupled or identical. This strongly suggests a close association of the two traits.

It is known that *A. cerana* workers remove *V. destructor*-infested worker brood with far greater efficiency than *A. mellifera* workers [11,41]. We show here that this is also true for *T. mercedesae*-infested brood. This may seem surprising given that brood removal is thought to be triggered by stimuli linked to the reproduction of mites on the brood [42], and that *T. mercedesae* only very rarely reproduces on *A. cerana* pupae [30]. An explanation could be that hygienic behavior is the reason for *Tropilaelaps* non-reproduction. Another explanation may lay in the fact that mite infestations are often lethal to *A. cerana* pupae [43], and that dead pupae presumably emit stronger cues for removal. Viral replication, which has also been reported to increase the likelihood of mite detection inside brood [44], is triggered by feeding of the parasite on its host, and so may also be involved in *T. mercedesae* recognition by *A. cerana*. In every case, the stronger hygienic behavior of *A. cerana* towards *T. mercedesae* offers an additional explanation for the fact that *A. cerana* colonies, in general, appear to be more resistant to mites of the genus *Tropilaelaps* than colonies of the western honey bee [26]. The relative importance of hygienic behavior with regard to other resistance mechanisms, including other mechanisms suppressing mite reproduction, will be an important subject of future research.

Our experiments with mixed groups of *A. cerana* and *A. mellifera* workers were originally designed to be able to directly compare the ethology and temporal pattern of hygienic behavior of the two species side by side on the same brood comb, i.e., in response to strictly the same stimuli. This turned out not to be possible, because hygienic behavior was almost completely suppressed in *A. cerana* workers by the presence of *A. mellifera* bees. This suppression was apparently not due to an inability of *A. cerana* workers to sense infestation in *A. mellifera* brood, because it equally occurred in mixed groups of bees on *A. cerana* brood, and because in the absence of *A. mellifera* workers, infested *A. mellifera* pupae were removed at a high proportion. This result is interesting with regard to the question of task affiliation between worker bees. In mixed colonies of *A. cerana* and *A. mellifera*, both species participate in almost equal numbers in queen retinue behavior [45], whereas *A. cerana* workers tend to specialize in foraging [46]. Communication via the dance language is possible across the species barrier [47]. One hypothesis to explain the suppression of brood hygiene in *A. cerana* is that *A. mellifera* workers possess a lower detection threshold for the signals emitted by infested pupae. Hygienic behavior is at least in part triggered by chemical cues that are perceived through the wax capping [48,49]. *A. cerana* pupae have been shown to react more strongly to mite infestations [43]. Therefore, it seems possible that a moderate olfactory sensitivity of *A. cerana* workers would be sufficient to reliably sense them, whereas *A. mellifera* may have had to evolve a greater sensitivity, because the stimuli emitted by infested pupae are weaker.

*T. mercedesae* has recently been detected on *A. mellifera* in several areas outside of the distribution range of *A. dorsata* [31]. Our study confirms that one of the central defense mechanisms of the western honey bee against brood diseases, hygienic behavior, is also expressed vis-à-vis this new parasitic mite. Given that hygienic behavior against *V. destructor* and *T. mercedesae* appear to be closely coupled, breeding for behavioral resistance to varroosis likely can reduce the vulnerability of *A. mellifera* beekeeping also to *T. mercedesae*, and possibly also to other species of this genus. This is of particular importance, given the increasing trend of *A. mellifera* colonies to produce brood throughout the winter, even in “temperate” regions, brought about by climate change. At the same time, greater efforts will be needed to elucidate the biology of the four *Tropilaelaps* species, where great knowledge gaps need to be filled e.g., with regard to reproduction biology, dispersal, and host–pathogen interactions.

## Figures and Tables

**Figure 1 insects-11-00510-f001:**
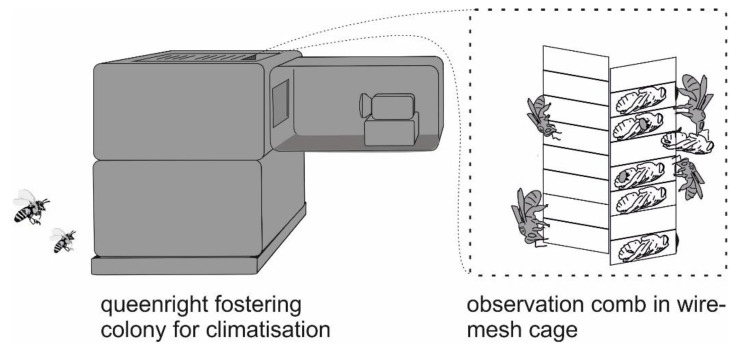
Experimental setup for the observation of hygienic behavior.

**Figure 2 insects-11-00510-f002:**
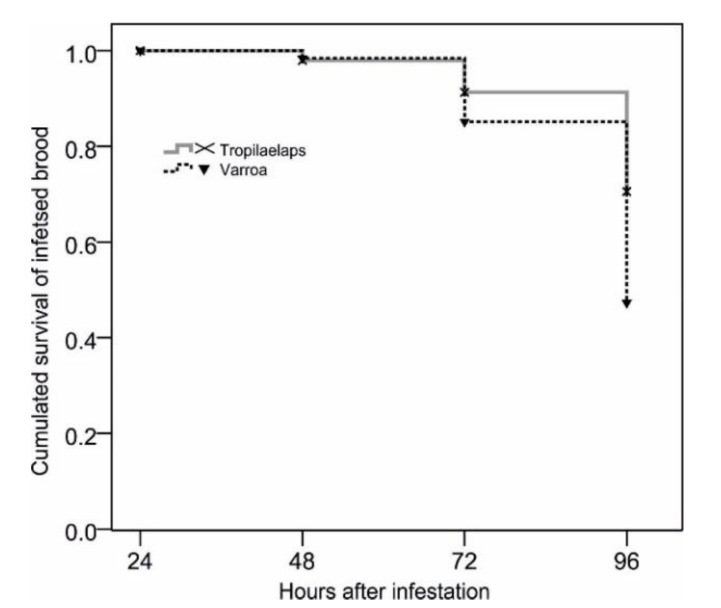
Survival of *Tropilaelaps mercedesae*- or *Varroa destructor*-infested brood by *Apis mellifera* worker bees.

**Figure 3 insects-11-00510-f003:**
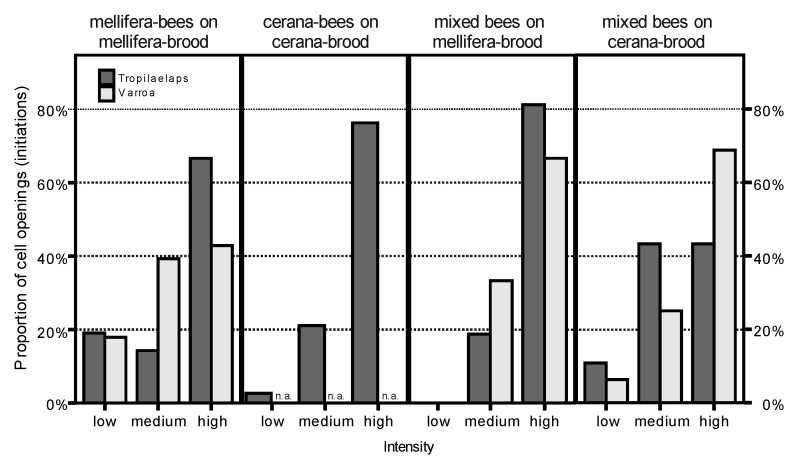
Intensity of cell opening behavior towards *V. destructor*- and *T. mercedesae*-infested brood.

**Figure 4 insects-11-00510-f004:**
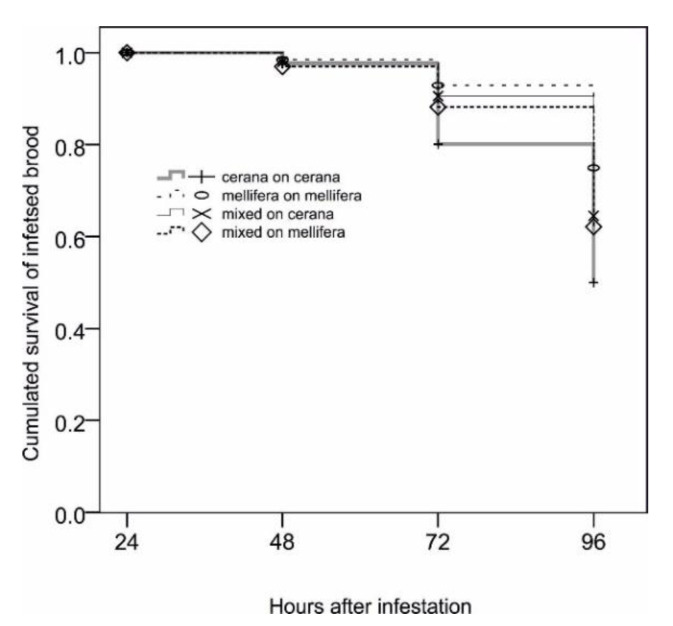
Survival of *T. mercedesae*-infested *Apis cerana* and *A. mellifera* brood in the presence of workers of both species.

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
