# Peer review of "Individual-Level Comparisons of Honey Bee (Hymenoptera: Apoidea) Hygienic Behavior Towards Brood Infested with *Varroa destructor* (Parasitiformes: Varroidae) or *Tropilaelaps mercedesae* (Mesostigmata: Laelapidae)"

_insects, 2020, doi:10.3390/insects11080510_

Round 1

Reviewer 1 Report

The work attempted in this study represents a novel and ambitious contribution to the scientific community but the execution of this study may obscure its usefulness. The parasitic mite Tropilaelaps mercedesae is considered to be of great concern to the global interests of apiculture and concomitant food security. Tropilaelaps has a pronounced effect on the viability of honey bee colonies and the potential for control via genetic resistance would be valuable. Genetic resistance already exists for the related mite, Varroa destructor, and this work seeks to determine if that Varroa resistance provides cross-resistance to Tropilaelaps. There are issues, however, both major and minor, with this manuscript that will need attention. These issues are matters of clarity, organization, and study design. Some issues may require resubmission with additional data or the removal of some data. Specifically, the lack of identification of haplotype in the mites used in this study cannot be ignored as it extends to the primary question to be tested in this study. The mites used in this work were all taken from A. mellifera but were introduced into the cells of Apis cerana. Some species of Varroa have been shown to be incapable of parasitizing A. cerana and their unnatural transplantation in an A. cerana colony may account for their rate of removal. Such unnatural circumstances could not then be extrapolated to a natural setting. For a study with a strong focus on the impact of genetics on the control of parasites, very little genetics work appears to be have been conducted as the issues arises again in the vague presentation of “ecotypes” of the host bees that are never further defined. More detailed analysis below:

Major Revisions:

Page 3, paragraph 1

Are these ecotypes distinct? They need to be described genetically if they’re to be used here.  

Page 3, paragraph 1

The author mentions 3 ecotypes but doesn’t discuss how many of each were used. This complex experimental design can only be reproduced by future researchers if the level of diversity in colonies is made known.

Further, was the rest of the brood checked for brood parasites? Treating the entire colony with an effective acaricide, confirming its effectiveness in reducing parasite loads below detectable levels and then conducting this work would make the most sense. It seems that the colonies were treated with an acaricide but no information is provided about the level of reduction in parasitic mite levels.

Page 3, paragraph 2

Which haplotypes of Varroa and Tropilaelaps were used in this study? The different haplotypes have different levels of virulence and could potentially prompt different responses from hygienic bees. All mites were collect from A. mellifera but were introduced to brood of two different species. This design doesn’t control for the fact that different haplotypes of Varroa and Tropilaelaps are apparently unable to or inefficient at parasitizing certain species of honey bees which could account for some elements of the data presented here. Not all work on Tropilaelaps or Varroa requires identification to the level of haplotype but an undertaking focused on identifying key genetic differences that may impact triggering of hygienic behaviors is certainly the sort of study where identification and disclosure of haplotype would be necessary. Speculation in the discussion references the potential impact of haplotype but does little to provide clarity about the data collected in this study.

Page 3, Section 2.3

Why were both species of bees placed in the same colony. One would expect the presence of a different species of honey bee to impact the behavior of the host colony potentially biasing the results. The data collected appears to show this was the case. The mixed species studies had a more than doubled rate of removal of unparasitized brood (sham manipulated) which may be a result of the general confusion caused by the unnatural set up in addition to the suppression of natural hygienic behavior in A. cerana. Justification of this choice in the methods or introduction is necessary to move forward as it is not encompassed in the primary hypothesis.

Page 5, Figure 2:

This figure is misleading as it makes it seems as if a very significant difference exists between the hygienic behavior of A. cerana when pressured by Varroa or Tropilaelaps as Varroa shows a total of zero uncapping incidents. However, Varroa were not used in the A. cerana on A. cerana trial. This graph should either be shown separately from the other graphs, the image itself should be made clearer (not just an addendum to the legend text), or the A. cerana on A. cerana data should be removed.

Minor Revisions:

Page 2, paragraph 1:

The premise that Tropilaelaps are unable to feed on adult bees is proposed but not proven. There is an absence of evidence supporting it which does not amount to evidence of absence. Please revise how you articulate this point.

Page 2, paragraph 1:

Referring to brood rearing as the "main condition" for T. mercedesae survival doesn't account for their continued survival in temperate regions of Korea and China where winters are harsh and brood rearing apparently ceases for several months on end. Thus, it is worth reevaluating whether it's accurate to call this "the main condition" Please revise.

Page 2, paragraph 3:

“Beekeepers working with A. mellifera in regions with any of the Tropilaelaps species have to find ways to control these parasites…” <This assertion is inaccurate given that only 2 of the 4 Tropilaelaps species (Tropilaelaps clareae and T. mercedesae) are known to parasitize A. mellifera (Anderson and Morgan 2007). Please revise.

Page 6, paragraph 1 :

It is not appropriate the speculate about the meaning of data in the results section. Please remove speculation such as “meaning that the two traits were coupled or identical” to the discussion section.

Author Response

Reactions to comments by reviewer 1

Nb.

Comment by reviewer

Reaction of authors

Pos. in revised manuscript

1

Specifically, the lack of identification of haplotype in the mites used in this study cannot be ignored as it extends to the primary question to be tested in this study.

.

We agree that haplotype analysis would have been desirable. However, the primary question addressed in this study was whether hygienic behaviours against Varroa- and Tropilaelaps-mites constitute the same trait in Apis mellifera. We addressed this question by checking whether the same specialized bees expressed both behaviours. The mites used were identified to the species-level as described in the methods-section, and we compared the removal rates of infested and sham-manipulated brood in order to make sure that hygienic behaviour was really triggered by the introduced mites. Unless one postulates that different haplotypes of Tropilaelaps mercedesae produce altogether different  brood removal stimuli in A. mellifera (for which, as far as we are aware, there is no evidence in the published literature), knowledge of haplotypes is not essential to answer our research question.

For the second research question, comparison of hygienic behaviour towards Tropilaelaps in cerana and mellifera-workers, virulence differences between haplotypes of T. would have been important, and we acknowledge this in the discussion.

However, there is to our knowledge no research showing differences in virulence of different haplotypes within the species T. mercedesae. Nepal is situated between two countries from which only the “Asian Mainland” haplotype has been reported, and so it is likely that this haplotype was also used in our study.

100-101

257-259

2

The mites used in this work were all taken from A. mellifera but were introduced into the cells of Apis cerana. Some species of Varroa have been shown to be incapable of parasitizing A. cerana and their unnatural transplantation in an A. cerana colony may account for their rate of removal. Such unnatural circumstances could not then be extrapolated to a natural setting

The Varroa-mites used here were identified by morphometrical cues to belong to the species V. destructor, which is definitely capable of parasitizing both Apis cerana and Apis mellifera (see for example Lekprayoon&al., 2005). The species V. destructor originates from A. cerana, and to our knowledge, all described haplotypes of V. destructor have been found to parasitize A. cerana. Therefore, it seems unlikely that the use of other Varroa destructor  haplotypes would have led to qualitatively different results. There are, however, reports of differential virulence of different haplotypes, and A. cerana and A. mellifera coexisting in one geographical region may be infested by different V. destructor haplotypes (e.g. Fuchs et al., 2000), and so quantitative differences may have been possible.  As determination of haplotypes was not possible within the framework of our study,  we have tried to at least add some information about the local haplotype of Varroa from the literature.

100-101

259-263

3

Page 3, paragraph 1: Are these ecotypes distinct? They need to be described genetically if they’re to be used here.

We have removed the term “ecotypes” and now merely mention the geographic origin of the bees used. 

95

4

The author mentions 3 ecotypes but doesn’t discuss how many of each were used. This complex experimental design can only be reproduced by future researchers if the level of diversity in colonies is made known.

The fact that bees of the two species were mixes of different stocks is now mentioned more explicitly.

96-97

5

Further, was the rest of the brood checked for brood parasites? Treating the entire colony with an effective acaricide, confirming its effectiveness in reducing parasite loads below detectable levels and then conducting this work would make the most sense. It seems that the colonies were treated with an acaricide but no information is provided about the level of reduction in parasitic mite levels.

The level of effectiveness of the miticide treatments was not determined directly, but the approximate level of infestation of the brood used for the experiment  was checked by opening 20-30 capped cells. This is now specified in the text.

105-108

6

Which haplotypes of Varroa and Tropilaelaps were used in this study? The different haplotypes have different levels of virulence and could potentially prompt different responses from hygienic bees. All mites were collect from A. mellifera but were introduced to brood of two different species. This design doesn’t control for the fact that different haplotypes of Varroa and Tropilaelaps are apparently unable to or inefficient at parasitizing certain species of honey bees which could account for some elements of the data presented here. Not all work on Tropilaelaps or Varroa requires identification to the level of haplotype but an undertaking focused on identifying key genetic differences that may impact triggering of hygienic behaviors is certainly the sort of study where identification and disclosure of haplotype would be necessary. Speculation in the discussion references the potential impact of haplotype but does little to provide clarity about the data collected in this study.

See answers 1. and 2.

7.

Why were both species of bees placed in the same colony. One would expect the presence of a different species of honey bee to impact the behavior of the host colony potentially biasing the results. The data collected appears to show this was the case. The mixed species studies had a more than doubled rate of removal of unparasitized brood (sham manipulated) which may be a result of the general confusion caused by the unnatural set up in addition to the suppression of natural hygienic behavior in A. cerana. Justification of this choice in the methods or introduction is necessary to move forward as it is not encompassed in the primary hypothesis.

The observation of mixed groups of mellifera- and cerana-bees was performed in order to expose both species of bees to strictly the same (the same individual pupae infested by the same individual mites), and compare their reactions. This is now specified in the introduction. Of course there was an a priori-risk that the presence of one species affecting the likelihood of hygienic behaviour of the other, and this turned out to be the case. We think that the results (mellifera-bees specializing in hygienic behaviour) are nevertheless worth being presented, because they lead to the interesting hypothesis of a higher response threshold in cerana-workers to infestation-related stimuli from brood. Reviewer 3 seems to share this view, hence his appraisal of the importance of our findings for the identification of removal cues specific to Mellifera and Cerana.

84-85

316-317

8.

Page 5, Figure 2:This figure is misleading as it makes it seems as if a very significant difference exists between the hygienic behavior of A. cerana when pressured by Varroa or Tropilaelaps as Varroa shows a total of zero uncapping incidents. However, Varroa were not used in the A. cerana on A. cerana trial. This graph should either be shown separately from the other graphs, the image itself should be made clearer (not just an addendum to the legend text), or the A. cerana on A. cerana data should be removed.

The title of Figure 2 is “Survival of T. mercedesae- or V. destructor-infested brood by A. mellifera-worker bees”. The graph shows only data obtained with Apis mellifera bees and brood. As can be seen from the graph, removal of Varroa-infested brood was not zero, but around 50% after 92 h.

We suspect that the comment by reviewer 1 aimed at figure 3, which depicts the intensity of the behaviour, not its frequency, and where indeed no data is shown for Varroa in the second plot because no cerana-brood was infested with this mite. We have added “n.a.” in the places where the bars would have appeared in order to increase the readability of the graph, and referenced this abbreviation in the caption. An revised eps-file is attached.

184

192, 198

Attached .eps

9

Page 2, paragraph 1:

The premise that Tropilaelaps are unable to feed on adult bees is proposed but not proven. There is an absence of evidence supporting it which does not amount to evidence of absence. Please revise how you articulate this point.

Changed; new wording is “It is not clear, however, whether”(…)

44

10

Referring to brood rearing as the "main condition" for T. mercedesae survival doesn't account for their continued survival in temperate regions of Korea and China where winters are harsh and brood rearing apparently ceases for several months on end. Thus, it is worth reevaluating whether it's accurate to call this "the main condition" Please revise.

Changed; new wording is  “an important factor”

46

11

“Beekeepers working with A. mellifera in regions with any of the Tropilaelaps species have to find ways to control these parasites…” <This assertion is inaccurate given that only 2 of the 4 Tropilaelaps species (Tropilaelaps clareae and T. mercedesae) are known to parasitize A. mellifera (Anderson and Morgan 2007). Please revise.

This is true. We have now added the information which two Tropilaelas-species are known to damage mellifera-colonies, and changed the sentence cited by the reviewer to “Beekeepers working with A. mellifera in regions where either T. mercedesae or T. clareae occur”.

59

12

It is not appropriate the speculate about the meaning of data in the results section. Please remove speculation such as “meaning that the two traits were coupled or identical” to the discussion section.

As a very similar statement was already contained in the discussion, the misplaced sentence in the results-section was simply removed.

209, 286

Reviewer 2 Report

Paper at hand offers new insight of the long-term hygienic behaviour research and raises the question is the hygienic behaviour towards V. destructor and T. mercedesae the same behaviour. Through the nicely designed experiment and well-made analysis of data, authors conclude that bees are able to equally remove brood cells infested with both mites. This give a nice and important conclusion that selective breeding towards hygienic removal of V. destructor infested brood cells will simultaneously increase hygienic removal of T. mercedesae infested brood cells, which is an important research nowadays when the spread of Tropilaelaps may be expected.

Introduction, results and discussion parts are written well and clear and I don’t have any major concerns.

Regarding Materials and Methods part, there are few questions/suggestions from my side. More specifically, regarding the parts of obtaining mites for infestation and artificial infestation of brood. You state that mites were collected from capped brood (pre-pupal to white-eye stage of development). At this time (white-eye stage), Varroa mite usually starts its reproduction cycle. On the other hand, it is described that freshly capped brood was taken for artificial infestation. Considering the facts that a) Varroa mites that started reproduction in one cell fail to reproduce if it is moved to other cell; b) the timing of infestation of brood cell affects the reproduction success of mite (see research of Kirrane et al. 2011) and c) the reproduction of Varroa mite may have an effect on hygienic removal of infested cell (VSH), are you completely sure that this didn’t affect the outcomes of your study? I advise you to highlight, if this is true, how the collected mites and methods of infection did not affect the reproduction ability of mites. Also, these problems should find few words in discussion part. Unfortunately, I am not familiar in details with reproduction cycle of T. mercedesae, but maybe this could explain a little why A. ceranae removed more T. mercedesae mites comparing to Varroa mites (if for example T. mercedesae mites are less affected by these manipulations).

Specific comment:

Short title – I suggest to follow the main title and put Varroa destructor before Tropilaelaps mercedesae.

Keywords - I advise adding Varroa destructor, Tropilaelaps mercedesae and hygienic behaviour.

Materials and methods

2.2. Bioassay - at the end of this section you describe intensity of behaviour as low, medium or high. Could you please provide in more detail what low, medium or high means?

References are not arranged according to the journal's instructions but I suppose this is left for the end with intent.

Author Response

Reactions to comments by reviewer 2

Nb.

Comment by reviewer

Reaction of authors

Pos. in revised manuscript

1

You state that mites were collected from capped brood (pre-pupal to white-eye stage of development). At this time (white-eye stage), Varroa mite usually starts its reproduction cycle. On the other hand, it is described that freshly capped brood was taken for artificial infestation. Considering the facts that a) Varroa mites that started reproduction in one cell fail to reproduce if it is moved to other cell; b) the timing of infestation of brood cell affects the reproduction success of mite (see research of Kirrane et al. 2011) and c) the reproduction of Varroa mite may have an effect on hygienic removal of infested cell (VSH), are you completely sure that this didn’t affect the outcomes of your study? I advise you to highlight, if this is true, how the collected mites and methods of infection did not affect the reproduction ability of mites. Also, these problems should find few words in discussion part.

We have used this bioassay for more than 20 years (e.g., Schöning et al., 2012; Bienefeld et al., 2015) and have validated it by checking that artificially introduced mites generally reproduce and manage to produce mature offspring. In most cases, mites are probably collected before reproduction starts. The description in the manuscript was slightly incorrect because mites for infestation are taken from brood of the stages freshly capped to white-eyed pupa, not prepupal to white eyed-pupa – this has now been corrected. However, it is true that in some cases, mite reproduction may be interrupted by the transfer. This problem is now acknowledged in the discussion. As the onset of reproduction occurs at a similar stage in Tropilaelaps and in Varroa, this is unlikely to influence the relative frequency/intensity of the behavioural response.   Also the main finding of the study, the fact that specialist workers opening cells infested by both mites, is not concerned by this problem.

263-270

2

Short title – I suggest to follow the main title and put Varroa destructor before Tropilaelaps mercedesae.

Changed according to suggestion

12

3

Keywords - I advise adding Varroa destructor, Tropilaelaps mercedesae and hygienic behaviour.

We leave the decision on this to the editor, because we do not know whether duplications of words from the title are acceptable as keywords.

4

Bioassay - at the end of this section you describe intensity of behaviour as low, medium or high. Could you please provide in more detail what low, medium or high means?

Short descriptions now added

129-130

Reviewer 3 Report

This paper shows results of well performed experiments using camera mediated direct observations of behaviour of individual bees. It also shows how difficult it can be te get high enough numbers of observations (so some cases too low to have significant results). However the main drawn conclusions are secure.

Nevertheless, I would have been very happy if one observation had been added: were the bees showing highest chance to both arrest Tropilaelaps as well as varroa, also the ones that responded most strongly to pin-killed or freeze killed brood? Here there would have been the chance to underpin whether or not mite (both Tr and V) infestation removal is associated strongly with these two assays often used in breeding.

Already suggested by the work of Page et al, it appears likely from your work that the difference in removal probability between A mellifera and A cerana is not based on higher sensitivity of a. cerana workers but more likely caused by higher damage of the pupae.

I appreciate the method to let a non hygienic colony repair the damage that might have  been caused by artificial infestation

Some small remarks:

  • page 2: Naturally infested may be written without a - in between (+ similar cases in the same paragraph)
  • the last sentence of P 4 appears to be a kind of heading? Same holds for P 6 after para 1: Comparison of ….This is not very clear
  • p 7 in Discussion: para 2 ((….traits such ...add as before VSH
  • same para: seem likely is twice so redundant: or: seem, or is likely

Good luck with the finalization of the paper.

Author Response

Reactions to comments by reviewer 3

Nb.

Comment by reviewer

Reaction of authors

Pos. in revised manuscript

1

Nevertheless, I would have been very happy if one observation had been added: were the bees showing highest chance to both arrest Tropilaelaps as well as varroa, also the ones that responded most strongly to pin-killed or freeze killed brood? Here there would have been the chance to underpin whether or not mite (both Tr and V) infestation removal is associated strongly with these two assays often used in breeding.

We agree that this is an interesting question. It may be worthwhile to perform a separate experiment on this. Within the present study however, the level of complexity, with two different mites and two different hosts, was already high, so it would not have been possible to include an additional factor such as “killed/infested”.

2

page 2: Naturally infested may be written without a - in between (+ similar cases in the same paragraph)

corrected

71-72

3

the last sentence of P 4 appears to be a kind of heading? Same holds for P 6 after para 1: Comparison of ….This is not very clear

Changed (by formatting as a heading)

184, 211, 235

4

·  p 7 in Discussion: para 2 ((….traits such ...add as before VSH

·  same para: seem likely is twice so redundant: or: seem, or is likely

corrected

272

281

Round 2

Reviewer 1 Report

The authors have made significant revisions to the manuscript and provided justifications for the methodology. Even without the identification of the parasites to haplotype, this work still represents a step forward in our understanding of TropilaelapsVarroa, and their hosts.

The inclusion of both species of bees in the same trial does provide interesting data. Further, I appreciate that the authors chose to include a trial that did not go planned as it is an important contribution to science that tends to be undervalued and left out of publication. 

I do think this study would have been stronger and more reproducible with the mites identified to haplotype and as the author points out, we do not know enough yet to know if haplotype would have any impact on the ability of the bees to successfully detect and remove the mites. However, the authors have provided sufficient justification for why they have chosen not to include that information.